# Drone Swarms as Networked Control Systems by Integration of Networking and Computing

**DOI:** 10.3390/s21082642

**Published:** 2021-04-09

**Authors:** Godwin Asaamoning, Paulo Mendes, Denis Rosário, Eduardo Cerqueira

**Affiliations:** 1COPELABS, Universidade Lusofóna, 1749-024 Lisbon, Portugal; 2Bolgatanga Technical University, Sumbrungu UB-0964-8505, Ghana; 3School of Communication, Architecture, Arts and Information Technologies, Universidade Lusofóna, 1749-024 Lisbon, Portugal; paulo.mendes@airbus.com; 4Airbus Central Research and Technology, Willy-Messerschmitt-Str 1, Taufkirchen, 82024 Munich, Germany; 5Computer Science Faculty, Federal University of Pará, Belém 66075-110, Brazil; denis@ufpa.br (D.R.); cerqueira@ufpa.br (E.C.)

**Keywords:** drone swarms, networked control systems, wireless networks, in-network computing

## Abstract

The study of multi-agent systems such as drone swarms has been intensified due to their cooperative behavior. Nonetheless, automating the control of a swarm is challenging as each drone operates under fluctuating wireless, networking and environment constraints. To tackle these challenges, we consider drone swarms as Networked Control Systems (NCS), where the control of the overall system is done enclosed within a wireless communication network. This is based on a tight interconnection between the networking and computational systems, aiming to efficiently support the basic control functionality, namely data collection and exchanging, decision-making, and the distribution of actuation commands. Based on a literature analysis, we do not find revision papers about design of drone swarms as NCS. In this review, we introduce an overview of how to develop self-organized drone swarms as NCS via the integration of a networking system and a computational system. In this sense, we describe the properties of the proposed components of a drone swarm as an NCS in terms of networking and computational systems. We also analyze their integration to increase the performance of a drone swarm. Finally, we identify a potential design choice, and a set of open research challenges for the integration of network and computing in a drone swarm as an NCS.

## 1. Introduction

Drones, otherwise called Unmanned Aerial Vehicles (UAVs), are autonomous or remotely piloted aircraft that have a large set of applicability scenarios, due to their versatility, low cost and easy deployment [1]. Applicability scenarios encompass both military and civilian areas [2]. In military deployments, drones are essential to salvage injuries and deaths caused on human pilots, as well as to perform surveillance recognition. There has been recently a rise in civilian applications including provision of wireless connectivity, collection of time-sensitive information from the ground, such as image and video, from disaster locations, delivery of goods and traffic control [3]. Much as drones are recognized as being useful in the range of scenarios, their individual use is challenged due to limited field of view and battery lifetime requirements for sustained tasks, such as covering large geographic areas and providing critical information on a timely basis [4].

To mitigate such limitations, several drones could collaborate with each other in a drone swarm to perform tasks more efficiently while providing coverage for large geographic areas [5]. This aim to develop swarms of drones have been tackled in projects such as the Low-cost UAV Swarm Technology project [6], demonstrated at the Farnborough air show in 2016, and the swarm technology introduced by the Chinese Electronics Technology Group [7].

Therefore, the development of drone swarms will increase the impact of drone systems based on the coordination of efforts among drones allowing for wider coverage, higher flexibility and robustness through redundancy [8]. In a swarm, each drone acts as a simple agent in a networked control system. All such agents are conjoined in a multi-agent system to work based on the integration of a networking system and a computational system aiming to allow simple behaviors to be distributed across all agents (drones), giving rise to collaborative behaviors capable of solving complex problems [9].

To tackle the technical challenges of drone swarms, researchers have been dedicating some attention to advances in wireless communications and in performance computation able to sustain for instance decentralized flight control operations [10]. However, most of these efforts aim to contribute to the creation of drone systems orchestrated through a centralized control, which to start with poses some scalability and security limitations. Moreover, in a wireless environment in which communication channels are subject to variations in its quality as well as potential malicious attacks, swarm functionalities such as collision avoidance and formation control may be jeopardized. In this scenario a drone swarm system will gain if developed based on self-organizing properties, thus requiring no reliance on any well-known controller, which would need frequent updates of status of the complete swarm system.

A successful operation of drone swarms in a self-organized manner requires the integration of the networking and computational systems [11], which until now have been mostly investigated in isolation. The self-organized operation of a drone swarm will rely on the tight consolidation of the networking and computational systems. Each of these two systems shall benefit from the presence of the other: for instance, the networking system should be able to find better transmission channels towards neighbors and/or better end-to-end paths towards faraway drones when a computational system estimates (or predicts if based on some artificial intelligence algorithm) that the quality (e.g., packet loss) of some wireless links may be endangered due for instance to shadowing effect, or some end-to-end path may become intermittently available, leading to a higher network robustness [12].

In this review, we intend to provide a state-of-the-art analysis of how to develop self-organized drone swarms as networked control systems via the integration of networking and computational system, including their coupling effects. In addition to literature overview, we also identify key research challenges and open issues to integrate networking and computing for developing drone swarms as networked control systems. To the best of our knowledge this is the first review that investigates the fundamental design choices to devise drone swarms as networked control systems via the integration of the networking and the computational systems. We summarize the main research contributions of this review as follows:(a)Analysis of prior efforts to develop drone swarms, in particular as networked control systems.(b)Analysis of two different types of design strategies to deploy drone swarms, namely non-interactive and interactive strategies.(c)Description of the basic functionality that needs to be provided by the networking and the computational systems.(d)Analysis of how the networking and computational system should be integrated to realize a networked control system.(e)Introduction to open issues related to the different natures of the networking and computing systems, as well as challenges related among others to the management of resources.

We start by providing a characterization of related research efforts in Section 2. Section 3 provides a description of how drone swarms look like as networked control systems, providing an analysis of two different deployment strategies. In Section 4 we provide an analysis of the basic functionalities expected from the networking and computational systems. Section 5 provides an analysis of how the networking and computational system should be integrated to realize a networked control system. Finally, in Section 6 and Section 7 we identify some challenges and open research issues, and we conclude this review.

## 2. Related Work

Despite all the applicability scenarios of drone swarms, there are still several development and deployment constraints that need to be solved to control drone swarms in an automated, secure and scalable manner. Aiming to tackle such research challenges, a first attempt passes by relying on a Software-Defined Networking (SDN) approach [13], where a centralized entity takes decisions about the operations of all drones in the swarm, being such drone deployed only gives enough functionalities to allow them to execute the directives sent by the SDN controller [5].

Such SDN approach is mostly suitable to deploy drone swarms based on a non-interacting strategy (c.f., Section 3.2). Prior research efforts such as Soft-RAN [14] and the SDN-WISE reported [15], proposed the design of the centralized control system by removing control decisions from drones, thus requiring the implementation of programmable interfaces to allow them to receive management intents from a network operator. One example is the usage of edge computing to push to the edge of a cellular network (e.g., collocated with a set of base station, or installed in a cloud radio access network) a set of control functions that normally require low latency from the request to the execution of an action derived from a computational process installed in the edge [16].

However, such approaches normally aim only to optimize networking functionalities, while they do not consider the dynamics of the drone swarm, such as network formation or the inter-dependencies between control of networking functionalities and computational functions such as flight control.

Besides centralized solutions, prior work already reports of some efforts to bring the computation not to the edge of the drone system, but to inside the drone system itself, more aligned with an interactive deployment strategy (c.f., Section 3.3). However, such efforts towards the distributed control of a drone system are usually conducted considering only the networking system [17] or the computational system [18].

However, the influences between the networking and the computational systems are not the focus of these surveys, although they can be instructive to deal with the problems from a cross-disciplinary perspective.

With the development of network architectures for drone swarms overly of interest today, there is the need to search methods that can handle the in-network mobility requirement and distributed computation functions across the network. However, in seeking to implement such efforts for swarm communication concepts, current IP-based solutions seem to be unable to satisfactorily handle mobility management. Due to that, many studies have recently implemented novel networking models aiming to provide support to couple networking frameworks for mobile networks with the capability to embed computation functions across the network. Some of those paradigms include Named Data Networking (NDN) [19], Information Centric Networking (ICN) [20] and Named Function Networking (NFN) paradigms [21], which may be capable of providing seamless connection support for mobility management on top of the existing IP-based communication framework at the networking layer. With such support the design of communication protocols for drone swarm can strive as such approaches are capable of performing data forwarding based on content locators and not host locators as happens with the traditional IP-based delivery of data [19]. Furthermore, enhancements were made to the NDN concept to develop the NFN network model, thus bringing on board the functionality to query the network about computation functions and not only to data objects. The NFN concept computes on data and has functionality to convert computed workflows into Interest packets to be forwarded in the network aiming to find a suitable data source or cache. Also, the NFN network controller is capable of querying the network to handle dynamic tasks service and to provide tasks resolution strategies while ICN and NDN are noted for effective handling of static tasks computations [22].

Therefore, based on these exhibited features of NFN, especially with regards to support for in-network execution of tasks, autonomous computation of functions as well as providing resolution for computing strategies and dynamic processes, the design of networking protocols, especially targeting control, computation and management of distributed networked systems, such as drone swarm may be effectively designed and implemented using NFN design concept. For instance, reference can be made of a Named Function as a Service (NFaaS) framework that is based on a NFN concept, allowing for dynamic performance of in-network execution of network resources [23]. The framework has the capability to couple networked nodes together while allowing for the deployment of forwarding strategies for message encompassing queries for network functions. Moreover, as happens with NDN, the NFN services are used to generate computational results extracted from data packets and disseminated by one data source to multiple clients on the network [24].

Another such implemented protocol framework is the Named Data Networking Function Chaining (NDN-FC) architecture, which seeks to embed information centric networking technology into Internet of Things (IoT) platforms [25]. The intent being to provide support for cut edging variety of functions based on merging Service Function chaining together with NDN, aiming to support in-network executions capable of facilitating the establishment of more resourceful IoT networks.

## 3. Drone Swarms as Networked Control Systems

Drone systems are mission-oriented in most cases. The missions might have different scales and complexities, requiring the deployment of networks with different strategies, as shown in Figure 1, Figure 2 and Figure 3. In a simple and low-scale mission, a drone system may encompass a single drone or a group of non-interacting drones, associated with a ground control center (c.f. Figure 1). For larger scale missions, multiple interactive drones, i.e., a drone swarm, may perform better through consensus and swarm intelligence (c.f. Figure 3). An overly complex system may encompass several drone swarms connected via a ground or satellite platform, via which they can share information. In such missions, a drone swarm may be modelled as a Networked Control System (NCS) [26].

### 3.1. Networked Control System Model

A drone swarm can be modelled as a NCS encompassing a set of computational systems (i.e., the drones) connected via a communication network; a NCS is a computational system that is controlled in a closed loop through a communication network. Specifically, the control and feedback messages are exchanged among the system computational units, or agents, in the form of information packet transmitted through a network. The functionality of a typical NCS is established using two basic elements: (i) a computational system able to gather data via sensors, to reach decision and to perform commands via actuators; (ii) a network, based on communication modules, standards and protocols (e.g., medium access control and routing) to enable the exchange of information.

Drones are equipped with on-board computers responsible for data processing, each of them can include several micro-services to perform specific tasks, such as sensing, decision-making (e.g., about the flight control), actuation on the environment or on the swarm itself [27]. Acting as a NCS the drone swarm can appraise the benefits derived from executions, by aiming at modifying its functioning, to realize the close-loop typical of NCS.

By evaluating and reacting, the effectiveness of the action of each drone is leveraged allowing them to behave as a group of multiple agents that work together aiming to produce a common goal; A swarm system exhibits emergent behaviors, where simple behaviors are shared across the many agents giving rise to a collective behavior capable of solving complex problems [28]. The deployment of drone swarms largely depends on the wireless and networking technologies for transmitting specific data and control commands among drones or between drones and the ground control, leading the deployment of drone swarms based on two strategies: non-interactive and interactive.

### 3.2. Non-Interactive Deployment Strategy

In a non-interactive deployment strategy, each drone is directly connected to a ground control station, which is used to monitor their status (e.g., location, condition of sensors, networking configurations), take decisions and send out new commands, such as new waypoints.

A non-interactive deployment strategy leads to a simple example of a network that cannot be further divided. This kind of strategy may be regarded as a single NCS considering that each drone can build a controlling closed loop to optimize its operation. The capability to communicate is also indispensable, since the drone still needs to interact with the ground control to complete its system operation of perceiving, communicating, computing and controlling.

#### 3.2.1. Network Topology

In a non-interactive deployment strategy, drones are controlled by the ground controller, meaning that this strategy relies on the existence of efficient uplinks and downlinks, as shown in Figure 1, allowing the ground controlled to transfer flight status to the drones and to collect data from them. In scenarios with multiple drones, drones are not capable of communicating directly nor even through the ground controller. Figure 1 illustrates a case with multiple drones.

In this type of simple deployment strategy there is a bidirectional communication link from each drone to the ground control system allowing for direct transmission of application specific data and control commands. This strategy is suitable for small size applicability scenarios with lower coverage range. The main limitation of this strategy is that the ground control station is a single point of failure on which the entire network can have a shutdown, together with transmission delays culminating from the long wireless links aiming to cover a wider area. From a communication point of view a non-interactive deployment strategy relies on a precise air-to-ground channel model, antenna design, and mobility model.

#### 3.2.2. Major Technologies

According to a non-interactive deployment strategy, drones gather relevant information from themselves and from the environment. The collected data needs to be processed, for which the drone needs to communicate with a ground control station. The result of the computational system in the ground station is communicated back to the drones, being used by the relevant local actuators.

From one hand, cellular technologies, such as LTE, can be used to provide improved system capacity, coverage, high data rates and reduced latency deployment [29]. For instance, LTE provides data rates delivery range between 0.07 to 1 Gbps with coverage range of up to 350 km on flexible spectrum. On the other hand, Low Power Wide-Area Network (LPWAN), such as LoRaWAN, provides broad area connectivity operating on unlicensed frequency bands with low data rate, power consumption, and throughput [30]. For instance, LoRaWAN provides data rates that vary from 0.3 kbps to 50 kbps [31].

In what concerns the computational system, this deployment strategy may rely simply on accurate mobility models or path planning functions based on intelligent algorithms, such as Particle Swarm Optimization (PSO), Ant Colony Optimization (ACO) and Bee Colony Optimization (BCO).

### 3.3. Interactive Deployment Strategy

In an interactive deployment strategy, drone swarms can be deployed in a completely autonomous manner, making their own decisions (e.g., about path adjustment) based on their perception of the environment including neighbor drones. In an interactive deployment strategy the drone swarm will be able to coordinate their operation based on a cooperative approach towards sensing, monitoring as well as information sharing, allowing them to get to a consensus about the best way to execute a mission while facing detected obstacles. In what concerns the exchange of information, an autonomous drone swarm must rely on a networking system that does not depend on the operation of a ground controller, with the exception of emergency situations.

In a full autonomous system, swarms rely on airborne networks with potentially strong line of sight attributes, which facilitate data transmissions, as well as with the capability to generate suitable routing tables allowing drones to take decisions based on a good understanding of the nearby neighbor. This localized operation, typical of self-organized systems, does not require a networking system capable of creating a full image of the complete drone swarm, which is an advantage in dynamic settings.

Independently of the considered network topology for the airborne network, the design of a reliable wireless communication system allowing for autonomous drone interaction under dynamic propagation conditions is essential for achieving efficient swarm operations [32]. Hence, an interactive deployment strategy leads to a more flexible swarm with higher density and range than a non-interactive strategy.

#### 3.3.1. Network Topology

The interactive deployment strategy requires drones to communicate with each other directly or through multi-hop links [33]. For that reason, the topology can be done with different levels of complexity from a network based on ground communication to several networks relying on air-to-air communication. In the latter case, the network can be simple, encompassing a single cluster, or more complex including multiple clusters or a large flat ad hoc network.

##### Infrastructure-Based Network

It follows the same topology as the non-interactive deployment (c.f. Figure 1), where all drones connect directly with a ground control station. The difference is that in an interactive deployment the drones can communicate among themselves via the ground station, which is responsible for relaying all communications. However, by relaying all control information via the ground controller, this strategy may lead to a low robust system, due to the existence of a single point of failure. In what concerns the decision-making process, this can be done in a centralized or decentralized manner. In the former case, all decisions are taken by the ground controlled based on information provided by all drones, In the latter case, each drone should be able to take their own decisions based on information that drones share among them via the ground controller, which in this case act only as relay.

##### Cluster-Based Network

All drones are connected to another drone that is set as configuration parameter during the deployment of the system or selected during operation based on the status of the swarm. Such drone is selected to operate as head of a cluster of neighbor drones, meaning that every communication generated or terminated in such neighbors are sent via the cluster head [34]. In this situation, the cluster head may become a bottleneck within the drone swarm causing link blockage and high latency. This system can be extended to include several clusters, each one with its own cluster head. In both cases (single or multiple clusters), the cluster heads are responsible to establish communication among the clusters and between each cluster and the ground control. The decision-making process can be done by the drones themselves in a distributed manner: based on individual decisions or in a consensus-based manner. This is possible due to drones being able to communicate with each other via cluster heads, and not only via the ground control. Figure 2 provides an illustration of a set of single clusters that forms a multi-cluster network.

##### Adhoc-Based Network

Drones can communicate via one-hop or multi-hop wireless network that normally has a flat organization, not relying on cluster heads [35]. Consequently, single node failures have little impact on the whole system. This deployment strategy brings reduced downlink bandwidth and latency requirements because of shorter links among drones. The advantages of an ad hoc-based network deployment strategy include network scale-up, fault tolerance, device autonomy, flexible and less expensive to setup [6].

In an ad hoc network setup the coverage of the drone swarm can be greatly bigger than in a cluster-based network due to the possibility to route packets over multiple hops [36]. However, since routing may not scale in a flat organization, an ad hoc network may be organized into many separate clusters, and each of them operating as a single ad hoc network. In this case, each ad hoc network will need to have one or several cluster heads, which will function as gateways. Figure 3 shows the overall topology of a set of ad hoc networks, which together might form a cluster-based ad hoc network.

#### 3.3.2. Major Technologies

To model a drone swarm as NCS, it is necessary to address the communication requirements capable of guaranteeing maximum system performance over an airborne wireless network. From a networking point of view, challenges of transmission failures, delays, fading channels and message errors may lead to degradation of the overall system performance.

It is essential to rely on wireless technology capable of connecting all drones together through effective mechanisms to allow for an effective interaction between drones [37]. Several technologies can be used to deploy an interactive strategy, including IEEE 802.11 [38], 3G/LTE [39], and satellite communications [40]. However, an alternative networking solution should be further investigated, since due to the mobility and power limitation of drones, the networking system needs to be able to operate even in the presence of irregular connection and device challenges, while still providing the needed delay and throughput requirements of several services. Moreover, wireless reliability and stability are key for the operation of swarms, but notably the communication protocols designed for close-loop control systems are challenged in providing the needed guarantees.

Such an alternative networking system should rely on routing protocols capable of providing secure data transmission routes from source to destination. In this context, literature already provides several studies about the different families of routing protocols suitable for drone effective communications. These protocols include Swam Intelligence-based protocols, Position-based routing protocols and Topology-based routing protocols [6].

In an interactive deployment strategy, computational resources are dispersed in different drones, as well as in the ground control station. This means that several computing tasks need to be designated to create a suitable decision-making process, namely to decide where and how the decisions are made. First, the coordination of computational tasks depends on the requirements of each one of them. For example, while simple flight control can be guaranteed by the controller on-board on each drone, more computing intensive tasks, such as image recognition may need to be transferred to the ground control, or to a set of drones that together may provide the needed resources. Moreover, the coordination of computational tasks may be done based on the exchanged information between drones, while avoiding links that may impact on the effective performance of the drone swarm.

## 4. System Components

Drone swarms encompass the integration of networking and computational systems, which together contribute to the closed-loop operation of the NCS. Looking into the full operational chain, sensing acts to gather data from the environment into the swarm system while the networking system ensures that such data gets to the points on the network (swarm) where computation should be done, as well as ensuring that computational results are delivered to the drones that require that information to take actions needed to adapt the overall behavior of the swarm. The remaining of this section analyzes the properties (basics, demands, challenges) of the two fundamental elements of a drone swarm: networking system and computational system.

### 4.1. Networking System

The networking system builds a communication graph for data to be exchanged between drones, as well as between drones and a ground infrastructure. The networking system should guarantee the closed loop by driving the data flow between sensing, decision-making and actuation. This is achieved by considering that those three elements (i.e., sensing, decision-making and actuation) may reside in different drones, providing a cooperative sensing. For instance, a drone may sense a new obstacle that does not interfere with its own position, but it may have an impact in the overall swarm. This information is made available to another drone able to compute an action, which may be executed by a set of other drones that are in the way of such obstacle.

In a drone swarm, communication is not only imperative for disseminating observations, tasks and control information, but can also assist in coordinating the drones more effectively and safely. The communication demands vary significantly in different applications. However, it is particularly a challenging task to provide robust networking, due to the energy limitation of drones and to external factors such as wireless shadowing and intermittent available links. The sub-sections below, are dedicated to analyzing the networking requirements and constraints as well as deliberations on fundamentals and pivotal properties that should characterize the networking system.

#### 4.1.1. Networking Demands

The data traffic related to the system operation can be classified into two major types, namely command and control traffic, as well as coordination traffic [41]. The former allows the ground control to monitor and influence the behavior of drones, as well as monitor messages with information about the drone’s status. The coordination of traffic encompasses information related to cooperation and collision avoidance. In addition to the mentioned system traffic, drone communication can also include payload data related to services implemented on-board such as data generated by observations of the physical environment (e.g., video cameras) and data generated and consumed by passengers.

In what concerns the air-to-ground communication links, the International Telecommunications Union (ITU), under The 3rd Generation Partnership Project (3GPP), classified command and control communication as well as payload communication for drone safe operations in terms of throughput, reliability and latency, as detailed in Table 1. These performance requirements aim to guarantee timely communication, processing and coordinated movement in real time. Although the information provided by 3GPP refers to air-to-ground links in a centralized scenario (c.f. Figure 1) the same quality requirements should be expected to any air-to-air links between drones.

Communication over air-to-air links is expected to have increased importance in autonomous drone systems, since drones require exchanged information among themselves to be able to take local decisions; Drone swarms require reliable wireless for real-time distributed coordination and processing to achieve system wide goals [43]. Hence, for swarms to attain a wide-area communication in support for coordinated and distributed real-time processing, the networking system needs to have network performance specifications embracing reliability, high throughput and low delays to offer greater cooperation and synchronization for effective control of aerial nodes [12].

#### 4.1.2. Networking Challenges

Mobility, including changing altitudes and speed is an inherent feature of a drone. Such mobility poses challenges in terms of reliability, due to the intermittent nature of wireless links and the different mobility patterns that drones may have during their mission. In this sense, the topology of a drone network remains unstable. Moreover, the different mobility patterns (direction and speed) of neighbor drones leads to challenges in terms of the doppler effect and difficulties in antenna alignment, which may lead to a decrease of network transmission performance. However, from another perspective the mobility of drones can be leveraged to enhance transmission and bandwidth capacity through the introduction of store-carry-and-forward (SCF) functions in the drones [44]. Hence, drones are subjected to several actuator failures and system uncertainties that impacts the system ability to sustain satisfactory performance. Uncertainties are inherent in the drone system itself, due to parametric variations of mass and inertia, velocity, altitude and position dynamics, elevation angles and random noise from the aircraft during flight.

Drone can be also subjected to external disturbance of weather and system uncertainties, namely related to the properties of the used wireless channels, such as large and small-scale fading. Large-scale and small-scale channel fading contributes to transmission reliability concerns and greatly influence the air-to-ground communication and operating performance of multi-UAV systems [45]. Large-scale fading usually accounts for challenges such as free-space fading, attenuation and shadowing [46]. Free-space fading phenomenon is reliant on LOS conditions as well as earth surface reflecting effects to decrease transmission strength as the waves propagate through free space [47]. Attenuation on the other hand, occurs due to loss in the radio signal transmission strength as it traverses in the frequency band [48]. The dominant cause of attenuation in air-to-air and air-to-ground transmissions come from atmospheric conditions of rain and gases. Rainfall mainly absorbs, scatter, and diffract wireless propagation into multiple paths thereby extending propagation paths of signals to be received in distorted manner from different paths and in so doing causing a rise in attenuation and transmission delays [49]. However, atmospheric gas molecules absorb the energy of propagating waves. Absorption is dependent on the temperature, pressure, altitude and the operating frequency band. Higher energy absorption from gases culminates into severe propagation attenuation [50]. On the other hand, shadowing effect can be caused by the body of UAV itself and misalignment due to the angular placement of antennas on the UAV [51]. Nonetheless, major constraint of shadowing effects in air-to-ground drone communication is largely attributed to restrictions with regards to acceptable operational altitudes of the drone, which limit its operations to lower altitudes and thus are inclined to shadowing impediments from high rising buildings and terrestrial objects on the LOS path of air-to-ground links to cause signal dispersion which impacts on communication efficiency [45]. Additionally, the authors mentioned that the frequency bands used for drone communication (c.f., 2.4 GHz, 3.4 GHz, and 5.8 GHz), are severely influenced by the environmental conditions mentioned including buildings, trees and distance between transmitter and receiver antennas to impact on the drone control systems ability to provide the needed coordination movement of the swarm. Meanwhile, thoughts of use of higher frequency carriers for drone propagation may result in high signal attenuation, leading to higher packet losses, as higher frequency bands have lower penetrating capabilities into obstacles when compared to low frequency carriers [48].

On one hand, small-scale fading results from reflection, scattering and diffraction effects emanating from constructive and destructive interference between wireless transmitter and a receiver, resulting in multi-path signals. The size, shape and texture of these objects defines the wireless channel characteristics [46]. Reflection could occur in air-to-ground channels of drones due to objects at the earth’s surface such as trees dispersing the propagation signals in a LOS channel into multi-paths [52]. Reflection can also occur from the airframe of the drone itself as well as from the flat segment of mountain slopes [53]. Diffraction occurs due to forest vegetation and foliage from various parts of trees such as trunk and top levels to impede the LOS air-to-ground propagation wave into multi-paths [54]. However, scattering of LOS paths of drones could be accredited to the composition of structures and objects located within the confines of the ground station as the drones operates at altitudes above ground levels [55]. Hence, drastic reflections, scattering and diffraction degrades propagation signal and may even result in complete loss of signals. Finally, mostly air-to-ground communications may potentially interfere with signals from other cells such as macro-cells and pico-cells thereby contributing to reduced channel capacities and throughput, which is referred to as interference [56].

#### 4.1.3. Networking Advances

After outlining the challenges faced by drone systems caused by wireless reliability issues, several actions can be taken to mitigate such challenges. Such actions aim to achieve wireless reliability with high throughput and low delays in support for better-quality channel propagation within the drone swarm.

##### Fading and Interference Mitigation

Drones may need to fly much higher above ground stations and high buildings, becoming more visible to multiple cells. On that note, the communication between drones and ground stations may potentially cause wave interference on other radio cells and vice versa due to a small angular spread by wireless [57]. There have been various studies towards alleviating inter-cell interference by using wireless beamforming, which is a prospective technology to enable communication under mmWave frequencies to effectively lessen inter-cell and inter-sector channel interference and fading, while extending transmission coverage for drone communication [58]. Beamforming adds other major benefits by increasing spectral and energy efficiency, as well as improving throughput. However, the design of such beam must avoid the creation of narrow beams and broad beams as their formation might incur into high cost on beam alignment training and extreme channel interference, respectively.

##### Efficient Antenna Configuration and Placement

The design, placement and configuration of antennas are crucial for the efficient exchange of control and application related data. Most critical for the selection of antenna type is the consideration for the high mobility nature of drones. Although omni-directional antennas are suitable for motion inclined tasks compared to directional antennas, it is expected that the received signal strength and throughput shall dwindle a bit. To alleviate this effect, the design of multiple dual-polarized and cross-polarized MIMO configured light-weight antenna systems of different characterization could be supportive. By this, appreciable number of antennas could be mounted at diverse alignments on the drone seeking to complement each antenna types of constraint during the drone maneuverings, to enhance signal quality and channel capacity in support for robust and high throughput channel propagation [59].

##### Multi-Hop Drone Cooperation

Due to the aerial mobility nature of drones together with their high operating altitudes, they could be exploited through multi-hop cooperation to serve as aerial base stations for improving wireless coverage efficiency, as well as used as multi-hop relays for capture and transfer of application data towards ground base stations [60]. Therefore, multi-hop drone design implementation shall yield to cease wireless constraints in shadowing, blockages and signal attenuation, to produce improved network spectrum efficiency with high throughput and reliability in support for real-time communication and operational needs of drone swarms. To that effect, we consider the goal of multi-hop drone cooperation in this work as an opportunity to explore the broadcast nature of wireless channels for cooperative forwarding opportunities for both wireless access extension and the operations of drone scenarios under high network throughput and reliability.

With regards to the intrinsic conditions of the drone’s system itself, which shows inefficiencies in actuator failures, system uncertainties and external disturbances during flights, a holistic control design is necessary to ensure for drone flight reliability, safety and fault tolerance to dynamic flying altitudes, position and stable trajectory for a system wide optimal performance [61]. Studies have shown that flight orientations, elevation angles, speed and direction are highly dependent on the wireless channel parameters, which requires for modelling channel parameters as a function of system orientation in the control design, while others are also proposing different approaches [42]. For instance, an investigation of propagation channel models for drones at flight heights of 0 to 100 m showed that the environment, heights of the adjoining obstacles, which causes reflections of the earth’s surface, together with the drone system uncertainties, all at varying heights, have adverse effects on the wireless propagation channels [62]. There are suitable control solutions from the literature to strike a balance for drone system uncertainties, actuator failures, misalignment and external disturbances. For instance, a finite-time fault-tolerant adaptive robust control for a class of uncertain nonlinear systems with saturation constraints using integral back-stepping approach [63], was proposed to achieve robust control and high-precision tracking performance for nonlinear systems such as robots, spacecraft and aircraft under actuator faults, external disturbance and system uncertainties. Another example is using distributed consensus tracking control for nonlinear multi-agent systems with state constraints [64] to control against multiple actuator faults, parametric uncertainties and external disturbance in quadrotors.

### 4.2. Computing System

Computation within a drone swarm relies in a process similar to a NCS in which the simple data may lead to complicated decision-making. Hence, the development of a suitable computational system relies on investigation of the best strategies for decision-making, depending upon the used deployment strategy and intelligence algorithms.

#### 4.2.1. Self-Organized Decision Making

In general, all drones have computation ability to achieve the basic flight control, but only the drones with high autonomy level can make decisions to fulfil the missions without interventions from a ground control station. In a drone swarm, the decision-making can be done by the drones themselves in a distributed manner, and by a centralized control entity. The latter occurs in swarms of drones with low autonomy. However, in a swarm of drones with high autonomy level, the decision-making can be done by the drones themselves.

Centralized decision-making process offers simple solutions in terms of the overall system design, while contributing to reduce the energy consumption of each drone. On the other hand, a distributed decision-making approach may lead to a more robust and scalable drone swarm. With this in mind, this review focus on distributed decision-making processes that may lead to more autonomous, robust and scalable drone swarm.

In more challenging applicability scenarios, it is expected drone swarms can operate in a self-organized manner. This means that drones need to collaborate, while facing fluctuating network conditions to maintain the network connectivity [12]. The self-organized nature of a drone swarm aims to assist distributed agents in each drone, in order to act by sharing the environment data, to observe conditions, and to autonomously decide on actions that will fulfil the system wide goals [65]. We present self-organizing distributed algorithms in Table 2 taking into consideration the algorithm name, function it performs and techniques deployed for your reference.

From the list of self-organizing distributed algorithms presented in Table 2, we reference one of the algorithms [69], to enable us explain control procedures used for implementing self-organizing decision-making controls for multi-UAV systems as shown in Figure 4. The control architecture is decentralized and decision-making by each individual UAV is autonomous, but their local interactions bring out their emergent global behavior. Network scale or removing a node does not impact network performance. For cooperation and function of each UAV, the self-organizing decision support procedures are put into three stages namely Perception, Decision and Action as shown in the Figure 4. Information on position and velocity of UAVs and neighbors are acquired by considering, stationary and mobile obstacles, after which processes decision-making-based on modelling and optimization are performed to improve outcomes. The actuator then processes the penultimate best possible decisions, and the processes iterate again.

To share system observations allowing for the emergence of a collective behavior, drones operation needs to also be synchronized. Synchronization allows for the coordination of events among individual agents in different drones to support their harmonious swarm operation [17]. It helps the distributed processors to have a common notion of time, through the exchange of beacon messages to allow for effective fault diagnosis and recovery. Synchronization facilitates to merge data from the different distributed nodes into a single meaningful information through what is called data fusion.

There are several techniques proposed in the literature aiming to provide various levels of synchronization functions in distributed systems. Mention can be made to decentralized techniques that exploit the resources of visible light communication to facilitate effective video streaming synchronization in coordinated digital cameras [76]. In what concerns systems that require a tight clock synchronization, an adaptive synchronization method has been proposed based on time slotted channel hopping technique to empower devices needing to synchronize the ability to track and predict its own clock trickling relative to neighbors [77]. Other approaches related to clock synchronization for the coordination of multi-robot models have been exploited, namely the theory of swamalators in space and time to form a coupling block for swarms [78].

Synchronization allows self-organized systems to be managed and coordinated aiming to response swiftly to internal and environmental influences on the system with regards to failing nodes, resource variability and other factors contributing towards the overall system objective. This demand is termed system adaptability, which mainly contribute for making drone swarms robust. Adaptability models are required to be instantiated at run-time to allow an effective system control while responding to varying mission demands, and topology changes due to mobility of drones [17].

The self-adaptation property of a swarm system relies on an efficient propagation of information within the swarms. This requires an efficient set of network functions capable of maintaining the global knowledge of the network while localizing the interactions between drones to ensure that they do not rely on global knowledge to operate. Such networking functions (e.g., routing) should support system scalability, allowing for the expansion of the network without hampering performance [17].

#### 4.2.2. Swarm Intelligence

The goal of swarm intelligence is to leverage the operation of drone systems to exhibit advanced and complex swarm behaviors through their cooperation, organization and information exchange. A swarm intelligence systems is made of a set of simple agents (drones in this case) interacting locally among themselves and with the environment. The operation of swarm intelligence systems aims to mimic some biological systems, where agents (e.g., ants, bees) follow simple rules which are not dictated by any central entity. In these systems the interactions among the agents give way to the emergence of global behaviors. In this context, swarm intelligence algorithms aim to control drone swarms in what concerns their behaviors based on two approaches namely optimization and consensus. This article aims to analyze optimization approaches such as PSO, ACO, and BCO [9], as well as consensus approaches mainly Paxos and Raft algorithms [79].

PSO proposed by Kennedy and Eberhart in 1995 is inspired by the flocking behavior of birds [9]. PSO can be used to allow drones to cooperate in searching for the best solution to solve an identified problem, such as avoiding an obstacle. For this purpose, the PSO algorithm continuously update of velocities and positions of the drones in the solution space. By sharing this information, the algorithm allows different drones to move towards best positions relative to their neighbors to maintain strong bonds for collective sailing in attaining optimum performance [80]. We shall present in the adjoining tables swarm intelligence algorithms implemented based on the two approaches. Table 3 references are made to optimization-based implemented algorithms, considering optimization approach, solutions provided in terms of function and deployed implemented technique.

The ACO algorithm is devised to assist in finding optimal paths based on the travelling behavior of ants in search for food sources while releasing pheromones on their paths [9]. As ants pursue food sources, the release of more pheromones on a path is a likely indication of a food source. The algorithm has strong cooperative behaviors and so has strong global optimal abilities as well as flexible to implement. It is applied in combinatorial optimization problems, job scheduling problems and network optimization routing [91].

The BCO algorithm was also devised to find optimal solutions to distributed control system problems such as scheduling, clustering and engineering designs taking inspiration from the collective intelligence of bee’s behavior in food search [92]. The difference towards other algorithms such as PSO and ACO is that the BCO algorithm relies on the different roles that bees have in a colony. Hence, this algorithm may have a good applicability to the optimization of swarms encompassing drones with different roles.

In what concerns consensus algorithms, the goal is to allow the drone swarm to work as a logical group capable of withstanding the failures of some of its members [79]. Within the consensus algorithms, Paxos [79] remains the foundation frontier of consensus building in distributed networked devices. Paxos performs optimization process by defining peer-to-peer consensus based on single majority decision and ensuring that only one result is agreed upon. Peer nodes can suggest, lead and share equal responsibilities to achieve a resolution. Paxos is also deployed as trade-off to fault tolerance in distributed systems.

As an alternative to provide harmonization in distributed systems, the Raft algorithm [79] aims to decompose the consensus problem through sub-groupings. The number of sub-groups to considered are clarified through determinism thus limiting inconsistent incidences between logs. In these sub-groups, elected leaders take responsibility to manage replicated logs, accept log entries from clients and replicate to other servers together with a notification of when to apply the log entries to their machines. In Table 4, we detail sample consensus algorithms implemented based on Paxos and Ralf models to provide harmonization in distributed systems.

## 5. Computing and Networking Integration

In drone swarms, the need to create a closed loop requires a tight coupling between the computational and networking systems, which implies mutual influences and dependencies. Therefore, understanding such consolidated system effects is important to be able to make the best usage of the scarce resources that each drone may have to tackle the sensing, computational and networking tasks.

### 5.1. Networking Supporting Computing

Intuitively, networking contributes to the computational system by setting up a network topology able to support the required data exchange between all drones in the swarm. The data required by the computational system is transported using the networking system, while the outcome of the computational system of each drone is usually shared with others through the networking system, in order to allow the overall system to reach consensus. Hence, the performance of the networking system may have a significant impact on the computational capability of the drone swarm, especially for tasks that require real-time data.

Based on a pure IP networking solution (e.g., typologies created and maintained by mobile ad hoc routing protocols), all drones need to know with which other drone they need to communicate in the process of a computational function (e.g., collision avoidance). This approach process leads to a scalability and performance problem, due to the amount of state that each drone needs to store, and the time needed to update that state. Hence, one potential solution passes by using a networking system based on named data objects [97], allowing each drone to fetch the required data needed for their computational function without having to know where that data is stored in the swarm system. Although a networking system able to name data objects, and not drones, may bring benefits in terms of system performance (e.g., low delays since needed data can be fetched from a cache deployed in a closer drone) it still requires a two-level operation: first all data needed for the computational function needs to be fetched; second the needed computational function (e.g., estimate an updated list of potential obstacles) needs to be discovered in the swarm system. In this case it is assumed that due to energy constraints, not all drones will be responsible to run all computational functions needed to control the overall swarm.

Although such network would be focused on providing a resolution system from service names to required data, a better solution may pass by a networking system able to resolve computational functions instead of names. To that end, Named Function Network (NFN) proposed [98], would be able to resolve computational functions while allowing the requester to be agnostic about the status of the drone swarm. This property contributes to the self-organization properties of a swarm, since drones will only need to store local information. However, the usage of a networking system based on NFN would treat the drone swarm as a fully trusted entity, which may not be the case. Hence, a NFN-based drone swarm should be capable of transparently evaluating the results of every computational function [24].

### 5.2. Computing Supporting Networking

The computational system may enhance the networking system in several situations; for instance, several environmental information, such as altitude or terrain changes may lead to changes in the used channel models, which will have an impact in the performance of the networking system. In this scenario, the networking performance can be improved by the computational system, since it can measure and model the wireless channel, and classify that channel according to predefined rules.

Several research attempts have been conducted in an attempt to embed computation into a networking system, aiming to improve the overall system performance. Instances of such intelligence algorithms (e.g., through P4-assisted constraint operations) [99] helps to mitigate the limitations brought by high latency and low transmission rates. These computational functions are normally focused on data plane operations, such as aggregation of acknowledgement messages, better support for multicast, improved routing decision, scalability of forwarding depending on current conditions (e.g., link quality, drone mobility), reduction of the communication overhead in the whole system considering communication costs constraints.

In a drone swarm, and due to the potential hazards brought by the usage of wireless channels, communication security and data protection are significant for a successful deployment of drone swarms. Misbehavior and information leakage can lead to physical damage and endanger the operation of the swarm. In this context embedding computing in the swarm would allow the processing of traffic and data directly in the network and at line-rate. Hence, an integration of computational and networking systems creates the needed support for providing the needed security and privacy mechanisms, namely: first by supporting a mechanism for preventing attacks and intrusion. Second for detecting intrusion and undesired behavior when it has already taken place. Third, embedding computing in the network augments the swarm capacity for analyzing potential incidents, preventing future attacks.

By introducing a computational system, the decision-making process can help drones to better control their networking considering their own properties as well as the status of the environment.

## 6. Open Research Issues

A first challenge comes from the need to integrate networking and computing, as mentioned in Section 5, due to the need to combine discrete computational with continuous networking processes. Mostly because in a drone swarm both the networking and the computational systems continuously try to estimate the best behavior that the swarm should have under specific conditions in strict spatial-temporal constraints. Moreover, modelling a drone swarm as a NCS needs to consider the influence of time in the overall system behavior, which is sometime neglected when looking at the system from an event model or agent model point of view. Alternatively, some networking functions are based on a best effort service that are not in line with the requirements of a NCS, namely the need for high reliability. Finally, drone swarms are usually distributed, which means that the system components may operate in different devices. Hence, when modelling these distributed systems, there are a set of fundamental properties that need to be included in the overall system, such as synchronization, handling timing and network state inconsistencies, and network delay.

A second challenge is related to the resource constraint nature of drones, when facing services that require high computational power, such as pattern recognition, and video processing. Therefore, there is the need to perform some computation offloading to address such challenges, by using novel paradigms such as mobile edge computing. For instance, game theory could be used together with resource scheduling and task allocation schemes to achieve a trade-off between computing time and energy consumption while offloading part of the computing task to a nearby ground infrastructure.

This leads to another challenge related to the management of resources in drone swarms. In a drone swarm resources are limited, meaning that the task of managing resources in the overall system may be done via cooperation to increase its efficiency. This scenario leads to the challenge of assigning tasks to various drones while trying to achieve parallel processing goals.

Among the resources in a drone, the energy is of special importance. In this aspect devising an efficient networking system is of importance since it consumes most of the energy when compared to the computational system. However, it would be of interest to extend the lifetime of the drone system and its capability to communicate. Hence, a challenge is related to the best way to orchestrate the networking and computational system to increase the lifetime of each drone and of the overall swarm. This orchestration process requires further cooperation among different drones, which is quite complicated when considering the motion characteristic of each drone and the tight coupling between the performance of the networking system and the overall behavior of the swarm.

Finally, when compared with traditional NCS, a drone swarm leads to increased security issues. First, drones operating in open spaces face a few limitations in what concerns potential access by third parties. Moreover, since the drone swarm is deployed based on a wireless communication infrastructure, its security is harder to achieve than when deploying a regular NCS based on a wired infrastructure. In this perspective, drone communication requires further investigation in what concerns security, since drones are more subjected to malicious attacks than a normal wired NCS. Section 5.1 highlights the benefits that embedding computing into the network may bring to tackle security challenges.

## 7. Conclusions

Drone swarms and NCSs are attracting great attention. Developing drone swarms as NCS is anticipated to assist in improving their performance when facing operational and environmental challenges. Hence, in that perspective this review contributes to a better understanding of the operation of drone swarms as NCSs. We started by providing an analysis of the topologies and technologies needed to implement two types of deployment strategies. This review also provided a description of the two building blocks of any drone swarm, the networking and computational systems, and a thorough analysis of how to integrate them to achieve a self-organized swarm system.

We end up showing that building a networking system that does not rely on identifying hosts (e.g., drones) but rather computational functions, which can be deployed in any drone, provides the baseline to tackle the major challenges identified for the development of drone swarms as networked control systems. Relying on a networking system able to resolve on-demand computation expressions composed from named data and functions in a transparent fashion to each drone, contributes to the self-organization properties of a swarm.

Based on the outcome of this paper, a next step would be the investigation of a service-aware networking framework, based on an evolution of the name function networking paradigm, were services are related to the operation of drone swarm namely intelligent swarming and consensus algorithms. Challenges include devising the best coupling of networking and computing, such as the selection of the best places (drones) to execute the needed computational operations, and how to support service chaining based on the dynamic properties of a swarm of drones. Other challenges include efficient resource discovery, enabling compute reuse while ensuring a secure scheme for computational offloading, when required. All these challenges will be considered in our future work. 

## Figures and Tables

**Figure 1 sensors-21-02642-f001:**
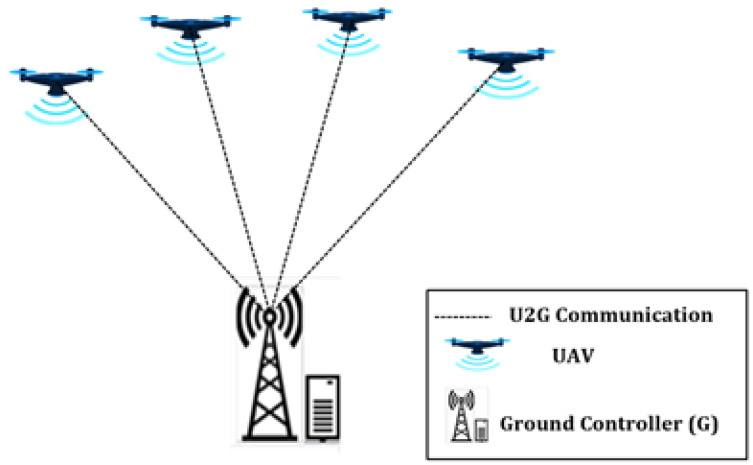
Non-interactive deployment.

**Figure 2 sensors-21-02642-f002:**
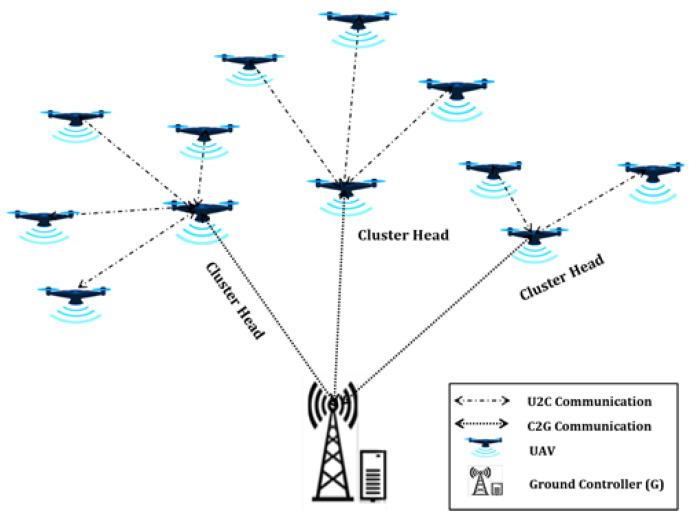
Multiple cluster interactive deployment.

**Figure 3 sensors-21-02642-f003:**
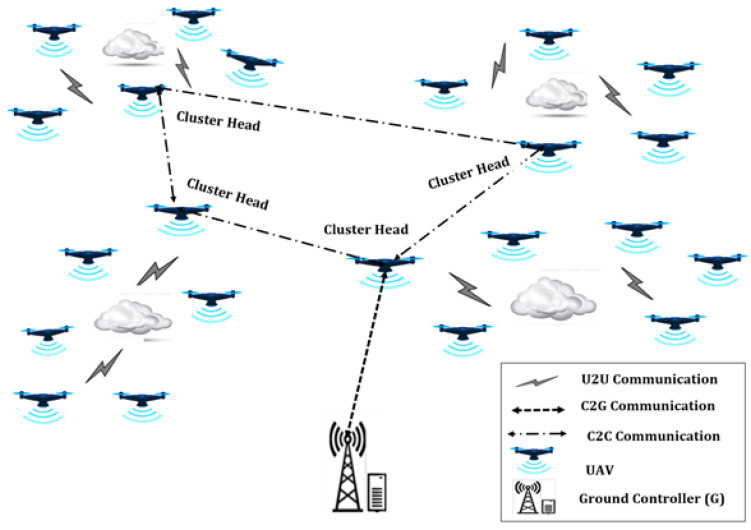
Cluster-based ad hoc interactive deployment.

**Figure 4 sensors-21-02642-f004:**
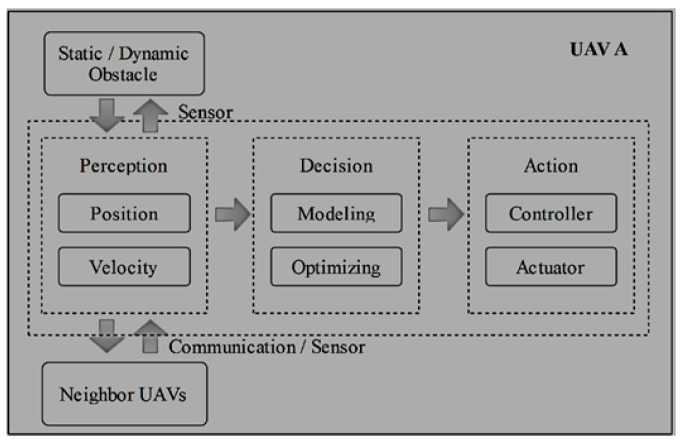
Self-Organizing Decision-Making Procedures.

**Table 1 sensors-21-02642-t001:** Air-to-ground communication requirements for drone operations specified by 3GPP [42].

	Data Type	Throughput	Reliability	Latency
DL (Ground station to UAV)	Command and control	60–100 Kbps	10${}^{-3}$ Packet error rate	50 ms
UL (UAV to Ground station)	Command and control	60–100 Kbps	10${}^{-3}$ Packet error rate	-
UL (UAV to Ground station)	Application Data	Up to 50 Mbps	-	Similar to Terrestrial user

**Table 2 sensors-21-02642-t002:** Self-Organizing distributed algorithms.

Ref.	Performed Function	Technique Deployed
[66]	Deployed to perform distributed cooperative search in Multi-UAV system	Implements the distributed decision making based on receding horizon techniques
[67]	Resolve slot access problem for neighbor UAV swarm network cooperation	Implements a collision discovery method to ensure slot access is not delayed by topology information exchanges
[68]	Implements self-organized collision avoidance in autonomous UAVs	Algorithm Compute’s safe reaction distance on which UAVs begin collision avoidance movement
[69]	Tackles sensing coverage constraints in multi-UAV swarm operations	Reciprocal decision-based approach performed between neighbor UAVs to reduce trajectory oscillations
[70]	Provides formation control for UAV and Mobile Robots communication networks	Distributed asymmetric control to implement formation control based on ‘mergeable nervous systems’ approach
[71]	Plays the role of implementing distributed control laws in swarm indexing and position-free density control	Pseudo-localization method used to localize agents to a new coordination frame with distributed control policies for desired coverage extensions
[72]	Solves limitations in transmission extensions in UAV wireless networks	Deploys relay selection techniques to improve coverage transmission constraints
[73]	Provides solution for slot access transmission problem in UAV Swarms	Makes use of game model as collision detection method to derive feedback in a common control channel for access slots
[74]	Performs flock distribution for UAVs to fly and coordinate as a unit	UAVs coordinates without a leader and on constant basis broadcast and as well receive movement information to share their common goal
[75]	Deployed to perform formation control in multi-UAV system	Formulates formation solution based on circle and arbitrary polygon formations techniques

**Table 3 sensors-21-02642-t003:** Optimization-based Algorithms.

Ref.	Optimization	Performed Function	Technique Deployed
[81]	PSO	Implemented path planning for swarms	Deploys jump-out and revisit methods to avert both null search attempts and local optimum
[82]	PSO	Implemented for finding moving targets using UAVs	Employs the Bayesian theory to convert a search problem to optimized cost function which will represent the probability of discovering targets
[83]	PSO	Implements cooperate path planning for Multi-UAV operations	Deploys time stamp model to manage UAV coordination expenses
[84]	PSO	Used to create exploratory trajectories for UAV networks	Delay tolerant networking approach is used for the team of UAV to follow
[85]	BCO	Implements UAV formation, obstacle avoidance control and target tracking	Deploys metaheuristic optimization approach exploited from the intelligent behavior of honeybee swarm
[86]	BCO	Implements flight planning solution for Multi-UAVs networks	Executed based on RNA coding procedure and so creates a coding technique pool to improve global search capability
[87]	BCO	Implements to achieve efficient Node localization process in UAV networks	Deploys UAV anchors to reduce localization oversights
[88]	ACO	Optimizes energy consumption and efficient path planning for UAV collision avoidance	Uses pheromone enhancement technique to implement a gain function for efficient path planning
[89]	ACO	Implements cooperative mission planning for UAV swarm target attack	Deploys time-sensitive target probability map for determining targets
[90]	ACO	Resolves cooperative search attacks mission planning in Multi-UAV networks	Uses distributed control architecture which separates the global optimization problem into separate sets to implement

**Table 4 sensors-21-02642-t004:** Consensus-based.

Ref.	Consensus	Performed Function	Technique Deployed
[93]	Paxos	Implemented to provides guaranteed consensus solution in distributed systems	Applies technique of flexible Paxos to reach consensus
[94]	Paxos	Provides a resolution to operational bottlenecks in consensus algorithms	Executes programmable network hardware to attain consensus service for requests
[95]	Raft	Solves inefficiencies and leader load balance issues, to enhance the stability of the algorithm	Uses Kbucket formed in the Kademlia protocol to influence the leader election process in the Raft algorithm, an approach seeking to improve the algorithm
[96]	Raft	Resolves message replication inconsistencies in blockchain networks	The Raft algorithm is enhanced through use of the Blockchain Hyperledger Fabric framework

## Data Availability

No applicable.

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
