# Peer review of "Drone Swarms as Networked Control Systems by Integration of Networking and Computing"

_sensors, 2021, doi:10.3390/s21082642_

Round 1
Reviewer 1 Report
- Interesting review manuscript well written. The list summarizing section 1 is particularly effective and will amplify the ease of reading.
- MAJOR ISSUE: The paper contains no quantitative analysis and no qualitative or quantitative results. Aside the breadth of the paper, the reviewer recommends publication as a Review, a Perspective, or a Communication rather than an original research article. Alternatively, theoretical and/or experimentally results can be added to the manuscript.
- Interesting bifurcation of the various approaches immediately followed by necessary technology.
- Please elaborate the respective content of doubly-cited [4,5] or reduce to the single relevant reference to reduce the burden upon the reader.
- Please elaborate the respective content of doubly-cited [44,45] or reduce to the single relevant reference to reduce the burden upon the reader.
Author Response
Thank you for reviewing our paper and providing valuable comments.
Please see the attachment for our point by point response.

Reviewer 2 Report
Paper deal with an interesting research area, unfortunately there is a lack of contribution. There is a lot of other scientific works dealing with same problem with better way.. There is also no strong scientific work on which the contribution is based..
no comparative info of another source..
paper sounds like a introduction part of the thesis rather than scientific article..
I found this paper not suitable for publication in your journal..
Author Response
Thank you very much for reviewing our paper and providing valuable comments.
Please see the attachment for our point-by-point response.

Reviewer 3 Report
The manuscript "Drone Swarms as Networked Control Systems by Integration of Networking and Computing" proposes to develop drone swarms as networked control systems based on a tight interconnection between the networking and computational systems, aiming to efficiently support the basic control functionality, namely data collection and exchanging, decision making, and the distribution of actuation commands.
Although the paper is well written and structured, I have some suggestions for the authors:
-please restructure the Abstract by removing the last part, from line 10, and add some information about the main results and the main contributions brought by the research to the field (a summary of lines 60-67;
-the discussion and results section is missing. please add it;
- please add the limitations of the study and the future paths.
The paper is very interesting to read and it is addressed to a wide range of audience. Also, as the authors declare, this is the first article that investigates the fundamental design choices to devise drone swarms as networked control systems via the integration of the networking and the computational systems.
Please make the above corrections and after that, in my opinion, the paper is suitable for being published within Sensors.
Author Response

(The authors gave the same response as above.)

Reviewer 4 Report
Dear Authors,
Reading the title and abstract, I expected a much more interesting article. The work does not present the authors' practical achievements, it is limited only to theoretical deliberations. However, the manuscript is a valuable literature review of the problem. I would be willing to recommend its publication, however, taking into account the following changes:
- the analysis should be extended to include the technical (state-of-the-art) possibilities of implementing individual concepts - taking into account the available communication standards and protocols,
- chapter 4.2 should be supplemented with a precise presentation of possible control and decision algorithms and the presentation of information flow diagrams in a swarm (distinguishing the type of information),
- figure 4 is understandable, but not attractive to the reader - the mentioned problems related to signal fading can be illustrated in a more interesting way,
- much of the information presented is fairly obvious or very general - additional drawings and mathematical relationships can make the work more attractive for the reader.
In terms of editing, the article is very well prepared.
Author Response

(The authors gave the same response as above.)

Round 2
Reviewer 2 Report
Dear Authors,
Thanks for all the corrections. I am impressed with the extent of the changes you have made to the manuscript. Overall, the quality of the article has increased significantly. i also appreciated change of the form to Review type.
Authors incorporated my suggestion.
Thus I suggest to accept article in present form.
Reviewer 3 Report
The authors answered to my comments and made the proper adjustments in the new version of the manuscript. Also, as the paper is a review paper, most aspects raised have been solved.
As such, in my opinion, the paper is now suitable for being published.
Reviewer 4 Report
Dear Authors,
After changing the manuscript type to Review and introducing almost all the suggested corrections, I decided to accept the manuscript as presented.
Best regards,
Reviewer